# Risk Perception Influence on Vaccination Program on COVID-19 in Chile: A Mathematical Model

**DOI:** 10.3390/ijerph19042022

**Published:** 2022-02-11

**Authors:** Juan Pablo Gutiérrez-Jara, Chiara Saracini

**Affiliations:** 1Centro de Investigación de Estudios Avanzados del Maule (CIEAM), Vicerrectoría de Investigación y Postgrado, Universidad Católica del Maule, Talca 3480112, Chile; 2Centro de Investigación en Neuropsicologia y Neurociencias Cognitivas (CINPSI Neurocog), Facultad de Ciencias de la Salud, Universidad Católica del Maule, Talca 3480112, Chile

**Keywords:** mathematical model, risk perception, vaccination, COVID-19

## Abstract

The SARS-CoV-2 virus emergency prompted unprecedented safety measures, which were accepted by the population of each country to different degrees, for example, with more or less willingness to use personal protective elements (PPEs). We have developed a mathematical model of the contagion process, based on chilean data, to assess the interaction between biological factors (such as the impact of vaccination) and behavioral factors (such as the population’s perception of risk). The model clearly shows that the virus spreads through three waves of contagion, the second being the most prominent, regardless of any alteration in the variables taken into account, which only affect the overall number of people infected. By considering alternative values of the risk perception variable and examining the different possible scenarios, we have also found that the less reaction to change the population has (and the lower the disposition to use PPEs), the higher the waves of contagion and the death toll are.

## 1. Introduction

Since December 2019, the world has been involved in a pandemic which has dramatically changed the way we consider a “normal” life. There have been other documented pandemics in the past (since the 1918 flu; [1]), but they did not have the impact that COVID-19 has had in modern life. It has involved practically all countries in the world and forced governments to take actions which had an important impact on the economical, political, and socio-cultural level [2].

With more than 200 million infected people and a death toll coming closer every day to 5 million as of this writing, the loss in terms of human lives and political/economic assets has been enormous, as the virus reduced global economic growth in 2020 to an annualized rate of −3.4% to −7.6% and global trade is estimated to have fallen by 5.3% [2]. Throughout 2020 and 2021, a number of countries have imposed significant restrictions to their citizens’ mobility and freedom, affecting the concept of workplace, social meetings, communication, arts and entertainment [3], and increasing the incidence of psychological or psychiatric issues [4,5,6,7,8].

Scientists from different areas and fields are still trying to investigate the multifaceted aspects of the phenomenon, in order to understand it and find a way to limit the number of casualties without undermining other aspects of human life. This way, some policies adopted by the governments of certain countries have been considered more “virtuous” than others, or more “effective” than others according to the citizens’ appraisal or epidemiological evaluation ([9,10]).

By constructing a mathematical model based on the way a phenomenon unfolds, we can find factors that can influence it through the variables associated with the model [11,12]. In particular, models related to COVID-19 have been typically used to determine the spread of the disease according to different factors that could influence the total number of infected, such as the mitigation measures presented by the affected countries and the epidemiological effect on population [13,14,15,16].

In classical mathematical models, behavioral variables, such as the reactions of individuals and their adaptation during the course of a pandemic are generally underrepresented. This information might modulate the outcome of classic models, bringing them closer to what happens in real life and, thus, offer more realistic information about pandemics. For example, a model proposed by Epstein and colleagues (2021) [17] considers a “triple contagion” model including a new variable, the vaccine, which fuels the “fear of contagion” variable with a new kind of fear: the “fear of the vaccination”.

Amongst the many countries pursuing an efficient vaccination process in order to reduce the contagion ([18]), Chile has been acknowledged as one of the countries that were able to lead a successful vaccination program, standing for some time at the third position amongst the countries with highest vaccination rates (after the UK and Israel [19]). Surprisingly, Chile also had to face a subsequent increase in contagions and death toll till the beginning of June 2021, when the new daily positive cases finally started to decrease.

In the present study, we developed a model using information shared by the WHO and by the Health Ministry of Chile (which mostly bases their knowledge base on the WHO indications), in their official page and in a public database [20]. As values for vaccine effectiveness on contagion prevention, hospitalization, ICU admission, and deaths, we have taken as a reference the study published from Jara and colleagues (2021) [21]. Their results show the infection rate of volunteers after a 3 months follow-up, and improve the accuracy of the effectiveness with respect to the first report.

However, how can the increase in contagion dynamics (waves, peaks amplitude, etc.) be explained as interacting with the vaccine effectiveness and other variables? Similarly to Epstein and colleagues [17], in fact, we observed that many people in Chile were “adjusting” their social behavior to the situation: when fewer infections were reported, they were more prone to go out, reduce social distancing, and neglecting personal protective equipment (PPE), and protocols (such as the use of face masks, alcohol, hands washing, etc.), while some other people did not change their attitude towards the pandemic at all (they never engaged social distancing or using personal protection items in any case), at the same time pushing forward a belief of vaccines as a “bad medicine”, refusing to receive their vaccination (more information on this topic [22]). On the other hand, there were also individuals who, having been vaccinated, felt an excessive sense of safety and stopped using PPE, social distancing, and other measures to prevent the spread altogether, which increased both their likelihood of spreading the virus amongst their non-vaccinated or partially-vaccinated contacts and their likelihood of becoming sick themselves and ending up in an ICU (since, according to Jara and Coll., 2021 [21], the vaccine effectiveness is, respectively, 65.9% against COVID-19 symptoms, 87.5% preventing hospitalization, 90.3% protective against ICU admission, and 86.3% avoiding COVID-19–related deaths). It is clear that the presence of these attitudes dramatically affects the output of any mathematical model describing the vaccine’s effect on the disease prevalence, and weakens efforts to present realistic scenarios in different countries. Therefore, in our model we included the possible adaptation (or lack thereof) of individuals to the epidemic context, observing how the model’s output changed by varying Risk Perception (we defined that variable as P) through Change Resistance Rate ($\Lambda_{1}$) and Reaction Speed ($\Lambda_{2}$). We have modeled these two variables considering results from previous pandemics (SARS) models as part of the more complex construct of Perception Risk (see for example Poletti et al., 2012 [23]) and following the model from Poletti et al., (2011; [24]) showing that spontaneous behavioral changes driven by cost/benefit considerations on the perceived risk of infection have an effect on the pandemic’s overall behavior. Risk Perception is a complex construct, which in pandemics is related to different factors applying to the general population (from mass media communication and sources of information [25]; to personal knowledge level [26]), although a small amount of people seem to be more reluctant to change their attitude [27]. In many cases, such as the Poletti and colleagues model or in the Bagnoli and colleagues (2007; [28]) the Risk Perception strictly depends on how fast people perceive the contagion is spreading around them.

Our first objective in this study was to observe how an increase in vaccine rollout speed would affect the situation in terms of infected people, casualties and hospitalizations and see if, for instance, during one year there would be fewer contagion peaks or fewer epidemic waves in case much more people were vaccinated. In this scenario, the Risk Perception variable has not been manipulated.

A second objective was to see if different scenarios where people had different behavioral reactions and Risk Perception levels (higher vs. lower) would have had an effect on the pandemic’s behavior (number of peaks, number of casualties, and people entering the ICU) independently from the vaccination rate, which, contrarily to the first model, in these two scenarios has been kept at a stable rate to isolate the variable of interest (that is, the behavioral change).

## 2. Materials and Methods

The mathematical model used to represent the COVID-19 dynamics has been built through a system of ordinary differential equations. We performed a variation to the classic compartmental model’s SIR (Susceptible (*S*)-Infectious (*I*)-Recovered (*R*)) from Kermack–McKendrick [29,30]. Because the COVID-19 infection has an incubation period, it is also advisable to assign the said compartment, which *E* will denote. After being infected, individuals may or may not present symptoms. To distinguish these two states, we denoted by *A* those asymptomatic and by *I* those with symptoms. Another factor to consider is people who need medical assistance after becoming infected; *T* denotes these. For the susceptible and recovered states, the notation S and R are maintained, respectively. In addition, after including vaccination, each state will be subdivided into two, those vaccinated and those without the vaccine, where those vaccinated will have a subscript *v*. The status summary can be seen in Table 1.

In the model dynamics, it is assumed that all individuals are born without disease at a rate *b*, and there is a mortality rate *d* that is not associated with the disease since the rate *m* is assigned for deaths from COVID-19. The infection occurs between the meeting of a susceptible and an infectious person through a β rate, which we differentiate between asymptomatic ($\beta_{a}$) and those who present symptoms ($\beta_{i}$). The average time during which an individual remains infectious is denoted by 1/α. Once the person leaves the incubation period, there are two alternatives; they present symptoms or not; $\lambda_{i}$ and $\lambda_{a}$, respectively, represent these proportions. Some individuals can be asymptomatic for a certain period of time and, subsequently, they present symptoms; we denote this flow by ϕ, while others acquire partial immunity at a $\gamma_{a}$ rate. Regarding individuals with symptoms (*I*), we propose two possibilities, one is that they need some medical treatment (*T*), and the other is that they recover by acquiring partial immunity (*R*); the rates associated with these two events are denoted by μ and $\gamma_{i}$, respectively. For individuals who need some medical treatment, either they recover ($\gamma_{T}$) or may eventually die from the disease (*m*). Additionally to the possible cases of reinfection, we have also considered partial immunity. Thus, the average time in which an individual remains immune is denoted by 1/ψ. The vaccination rate is given by *f*, while $f_{v}$ gives the total loss of the effect of the vaccine. The rates associated with vaccinated people differ from those not immunized by the *v* index.

Other factors to consider are those that reduce the susceptibility and infectivity of vaccinated people. These factors will be denoted by σ and δ, respectively. One aspect to highlight is the perception of risk (*P*) of people towards the COVID-19 pandemic. Based on [31,32], the associated equation is expressed by $P=\Lambda_{1}\left(P-P^{*}\right)+\Lambda_{2}\left(I+I_{v}+T+T_{v}\right)$, where $\Lambda_{1}$ is defined as the rate of resistance to change, $\Lambda_{2}$ the reaction speed, and $P^{*}$ the perception of the quantified average risk. Thus, we have defined the rate of transition β as dependent on the Risk Perception ($\beta =\beta^{*}*P^{*}/P$). It should be noted that the Risk Perception variable has certain limitations in this model, one of which is not considering certain factors of social behavior, prior knowledge, feelings about the problem, among others; but these variables were not an object of investigation here. The summary of parameters is observed from Table 2, and the flow diagram representing the dynamics of the disease is observed in Figure 1.

The model is expressed by the following system of differential equations:(1)$$\left\{\begin{array}{c} \begin{array}{ccc} S^{\prime } & =bN-S\left[\left(\beta_{a}A+\beta_{i}I\right)/N_{u}+\delta \left(\beta_{a}A_{v}+\beta_{i}I_{v}\right)/N_{v}\right]+\psi R-\left(f+d\right)S+f_{v}S_{v}\\ S_{v}^{\prime } & =fS-\sigma S_{v}\left[\left(\beta_{a}^{v}A+\beta_{i}^{v}I\right)/N_{v}+\delta \left(\beta_{a}^{v}A_{v}+\beta_{i}^{v}I_{v}\right)/N_{v}\right]+\psi_{v}R_{v}-\left(f_{v}+d\right)S_{v}\\ E^{\prime } & =S\left[\left(\beta_{a}A+\beta_{i}I\right)/N_{u}+\delta \left(\beta_{a}A_{v}+\beta_{i}I_{v}\right)/N_{u}\right]-\left(\alpha +d\right)E\\ E_{v}^{\prime } & =\sigma S_{v}\left[\left(\beta_{a}^{v}A+\beta_{i}^{v}I\right)/N_{u}+\delta \left(\beta_{a}^{v}A_{v}+\beta_{i}^{v}I_{v}\right)/N_{v}\right]-\left(\alpha_{v}+d\right)E_{v}\\ A^{\prime } & =\lambda_{a}\alpha E-\left(\phi +\gamma_{a}+d\right)A\\ A_{v}^{\prime } & =\lambda_{a}^{v}\alpha_{v}E_{v}-\left(\phi_{v}+\gamma_{a}^{v}+d\right)A_{v}\\ I^{\prime } & =\lambda_{i}\alpha E+\phi A-\left(\mu +\gamma_{i}+d\right)I\\ I_{v}^{\prime } & =\lambda_{i}^{v}\alpha_{v}E_{v}+\phi_{v}A_{v}-\left(\mu_{v}+\gamma_{i}^{v}+d\right)I_{v}\\ T^{\prime } & =\mu I-(\gamma_{T}+d+m)T\\ T_{v}^{\prime } & =\mu_{v}I_{v}-\left(\gamma_{T}^{v}+d+m_{v}\right)T_{v}\\ R^{\prime } & =\gamma_{a}A+\gamma_{i}I+\gamma_{T}T-\left(\psi +d\right)R\\ R_{v}^{\prime } & =\gamma_{a}^{v}A_{v}+\gamma_{i}^{v}I_{v}+\gamma_{T}^{v}T_{v}-\left(\psi_{v}+d\right)R_{v}\\ \; & \; & \\ P^{\prime } & =-\Lambda_{1}\left(P-P^{*}\right)+\Lambda_{2}\left(I+I_{v}+T+T_{v}\right)\\ P_{v}^{\prime } & =-\Lambda_{1}^{v}\left(P_{v}-P_{v}^{*}\right)+\Lambda_{2}^{v}\left(I+I_{v}+T+T_{v}\right) \end{array} \end{array}\right.$$

$\beta_{x}=\beta_{x}^{*}\frac{P^{*}}{P}$ and $\beta_{x}^{v}=\beta_{x}^{*}\frac{P_{v}^{*}}{P_{v}}$. Observe from (Equation 1) that by increasing the perception of risk, the probability of contagion decreases.

## 3. Results

The numerical simulations associated with the proposed Model (Equation 1) were performed using Matlab software [33], in particular the “ode45” function to solve nonstiff differential equations. The baseline values of the parameters related to the different rates are those mentioned in Appendix A (see Appendix A). Figure 2 shows the dynamics of the general infected population (black line), that is, those with or no symptoms, including those that need medical intervention, differentiating the vaccinated (blue line) from the unvaccinated (red line). During the 360 simulated days, three “waves” of contagion are seen, where the second reaches a significant peak. After incorporating vaccination, apart from reducing the quantity of infections, it is possible to observe a slight delay in the spreading of the contagion with respect to the unvaccinated population.

One of the main concerns after the onset of the pandemic is the number of people who need medical assistance, since the resources to provide it are limited. From Figure 3, it can be seen that after increasing the number of daily vaccinations (Figure 3b,c), there is only one significant “wave” of cases that require medical intervention, and mainly in the unvaccinated population, unlike the base case. (Figure 3a). Another aspect to highlight is that the “wave” happens earlier, which may be associated with the fact that the first wave disappears as there is growth until the second peak, producing a unification between the waves. After this, there is a significant flattening of the curve after increasing the daily vaccination dose. Finally, after increasing the daily dose, it is also observed that the trend of cases at the end of the period is reduced by half (see Figure 3).

Another factor associated with the development of pandemics is people’s perception of risk. This factor from our model (Equation 1) is affected mainly by resistance to change ($\Lambda_{1}$) and reaction speed ($\Lambda_{2}$). From Figure 4 and Figure 5, it can be observed that, after reducing the resistance rate by half the change, there is a decrease in the peak of people who need medical intervention, and the third “wave” flattens considerably (see Figure 4b and Figure 5b). Moreover, adding a faster reaction from people (obtained by doubling the values of reaction speed), the third “wave” disappears, and the peak of the second also decreases (see Figure 4c and Figure 5c).

Finally, the greatest threat posed by this pandemic is the number of deaths attributable to this disease. After comparing the accumulated ends of the base case with the variation in risk perception, we can show how people’s behavior brings to a scenario with a reduced number of deaths (see Figure 6). If the resistance to change decreases by half, the number of deaths decreases by approximately 16% (see Figure 6b); and if we change the values simulating that people additionally to that react faster (that is, doubling the reaction speed), the number of deaths decreases by approximately 31% (see Figure 6c) compared to the base case (see Figure 6a).

## 4. Discussion and Conclusions

We propose a COVID-19 contagion model that considers variables that are generally not included in mathematical models when accounting for vaccination modeling. In particular, the Risk Perception (*P*) and the two associated factors, the Reaction Speed of the population to adopt PPEs and behavioral preventive measures ($\Lambda_{2}$), and the rate of Resistance to Change ($\Lambda_{1}$); the modeling of these two factors has been inspired by previous models about SARS pandemic [23,24,28], where spontaneous behavioral changes in people are driven by the perceived risk of infection which, of course, can affect the pandemic’s overall behavior. In our case, we modeled the resistance to change factor through a time factor (the faster the reaction, the more the willingness to adapt to change one’s behavior). Another important factor that we added in our model is the possibility of being re-infected or susceptible to the virus despite being fully vaccinated (that is, after 2 weeks of receiving the 2 doses of CoronaVac shots in Chile), which has been observed during the COVID-19 pandemic and we thought was missing in previous models. Since the objective of this study was not to evaluate the impact of personal beliefs about vaccination on infection, we decided to initiate the numerical values associated with the perception of risk without distinguishing between vaccinated and unvaccinated ($P^{*}=P_{v}^{*}$, $\Lambda_{1}=\Lambda_{1}^{v}$, and $\Lambda_{2}=\Lambda_{2}^{v}$). In future studies, this variable could be manipulated to observe the impact of individual’s beliefs and attitudes toward vaccination (pro-vax or no-vax).

The proposed vaccination model’s novelty is that, in the equation, the transmission rate depended also on Risk Perception, which varies over time and responds to internal values variations of both number of daily cases and patients entering the ICU needing treatment. This led to obtaining an approximation to reality, via numerical simulations, regarding the number of waves that occurred in some countries where the pandemic has hit. Our model has some limitations, but can still give hints about the importance of the modeled variables: although the model does not consider essential variables in the spread of the disease, such as mitigation measures, access to vaccines, socioeconomic status, and mutations of the virus, it shows that the behavioral variable (Risk Perception) is essential for controlling this disease. If this variable is supported with mitigation measures, good access to vaccines, and others more to consider, peak infections would be better controlled. Therefore, developing mechanisms to increase Risk Perception is a factor that should be considered to enhance mitigation measures.

Our model shows that, regardless of the vaccination rate or the risk perception and reactivity of the population, the contagion goes trough 3 waves (with the second one bigger than the others) over the course of one year (360 days). Not only the non-vaccinated people risk more contagion than people who have been fully vaccinated, but also they are more likely to need health care and eventually die. This is more or less what we could see in different countries [34], where there have been 3 waves of contagion (the second one with more infected people), independently from the politics actuated from local governments and vaccination rates.

In Chile, a third shot is going to be performed, and this will probably change the scenario of our model, but still we believe that the model was able to capture the dynamics of the pandemic considering a very important variable, which is people’s behavior.

## Figures and Tables

**Figure 1 ijerph-19-02022-f001:**
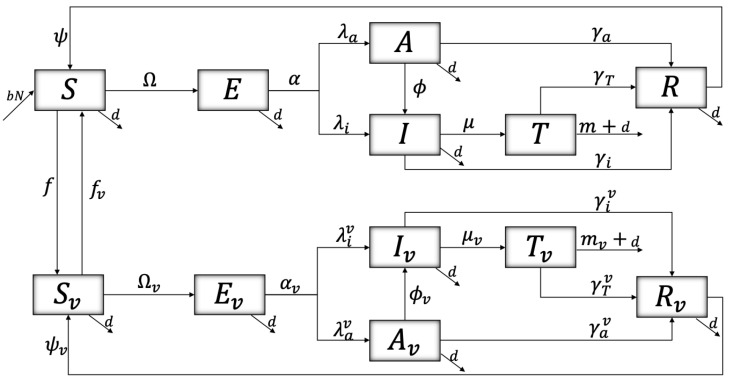
The model is divided into two main groups: (i) the unvaccinated and (ii) the vaccinated, differentiating (ii) from (i) by their *v* index. The general dynamics of the two groups are similar, except for the associated values at the respective rates. It is assumed that all entering the model are unvaccinated susceptible. Susceptible people (*S* and $S_{V}$) are infected after encountering an infectious individual, either with symptoms (*I* and $I_{v}$) or without them (*A* and $A_{v}$), differing mainly by the respective transmission rates. Once the infection is contracted, it has an incubation period for some time (*E* or $E_{v}$) to become later infectious (*A*, *I*, $A_{v}$, or $I_{v}$). Asymptomatic people, after a while, may have symptoms. There are individuals with symptoms that need medical intervention (*T* or $T_{v}$), and some of them might die from the disease (*m* or $m_{v}$). The recovery of people, whether asymptomatic, with symptoms and of those who need medical intervention, are directed to the *R* or $R_{v}$ compartment depending on whether (i) or (ii), whose immunity is temporary, becoming susceptible again. Note that $\Omega =\left(\beta_{a}A+\beta_{i}I\right)/N_{u}+\delta \left(\beta_{a}A_{v}+\beta_{i}I_{v}\right)/N_{v}$, $\Omega_{v}=\left(\beta_{a}^{v}A+\beta_{i}^{v}I\right)/N_{u}+\delta \left(\beta_{a}^{v}A_{v}+\beta_{i}^{v}I_{v}\right)/N_{u}$, where $N_{v}$ and $N_{u}$ correspond to the vaccinated and unvaccinated populations, respectively. The transmission rate is dependent on the risk perception (*P*) so that $\beta_{x}=\beta_{x}^{*}\frac{P^{*}}{P}$ and $\beta_{x}^{v}=\beta_{x}^{*}\frac{P_{v}^{*}}{P_{v}}$, with x∈{a,i}. $N=N_{u}+N_{v}$.

**Figure 2 ijerph-19-02022-f002:**
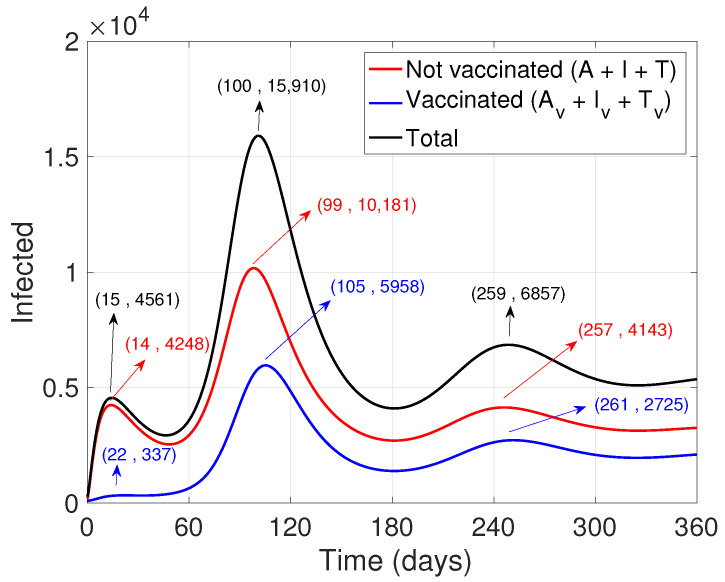
Dynamics of the vaccinated and unvaccinated infected, that is, adding asymptomatic patients, with symptoms, and who are undergoing health treatment. The initial values are S(0)= 99,000, $S_{v}\left(0\right)=100$, E(0)=500, $E_{v}=100$, A(0)=100, $A_{v}\left(0\right)=85$, I(0)=100, $I_{v}\left(0\right)=10$, T(0)=0, $T_{v}\left(0\right)=0$, R(0)=5, $R_{v}\left(0\right)=0$, $P\left(0\right)=P^{*}$, and $P_{v}\left(0\right)=P_{v}^{*}$.

**Figure 3 ijerph-19-02022-f003:**
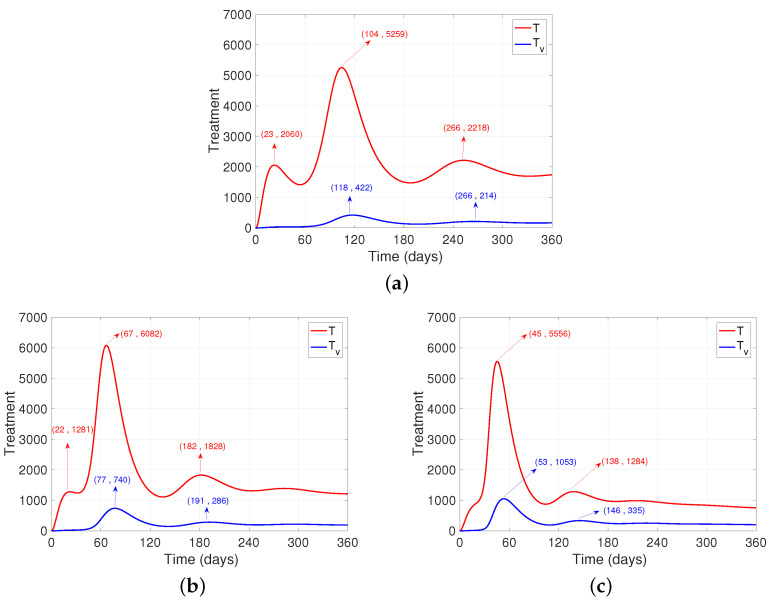
Dynamics of people who need some intervention after varying the vaccination rate. It is observed for the unvaccinated (*T*) and the vaccinated ($T_{v}$). (**a**) Base case. (**b**) Application of the double dose of daily vaccines. (**c**) Quadruplication of daily vaccine doses.

**Figure 4 ijerph-19-02022-f004:**
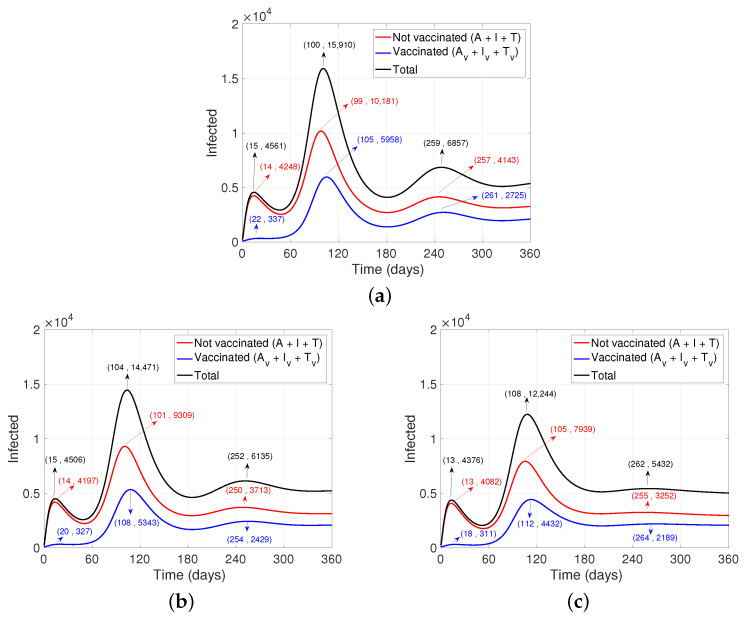
Dynamics of the behavior of the pandemic after varying the perception of risk baseline values. (**a**) Base case. (**b**) Halving of the rate of resistance to change ($\Lambda_{1}$), that is, people are less reluctant to behavioral change. A similar graph is obtained after doubling the speed of people’s reaction to the pandemic ($\Lambda_{2}$). (**c**) Reduction of resistance to change by half and increase in reaction speed by double.

**Figure 5 ijerph-19-02022-f005:**
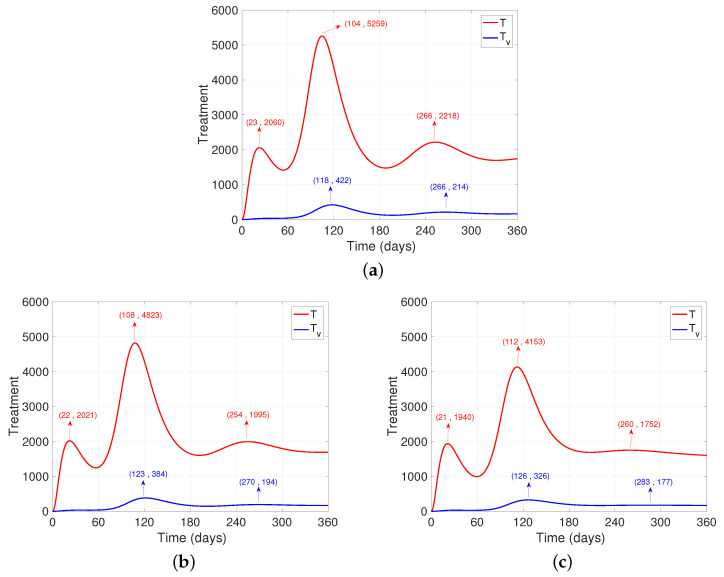
Dynamics of people who need some intervention after varying the baseline values of Risk Perception. (**a**) Base case. (**b**) Halving of the rate of resistance to change ($\Lambda_{1}$), that is, people are less reticent to change their behavior and start adopting PPE and reduce social distancing. A similar graph is obtained after doubling the speed of people’s reaction to the pandemic ($\Lambda_{2}$). (**c**) Reduction of resistance to change by half and increase in reaction speed by double.

**Figure 6 ijerph-19-02022-f006:**
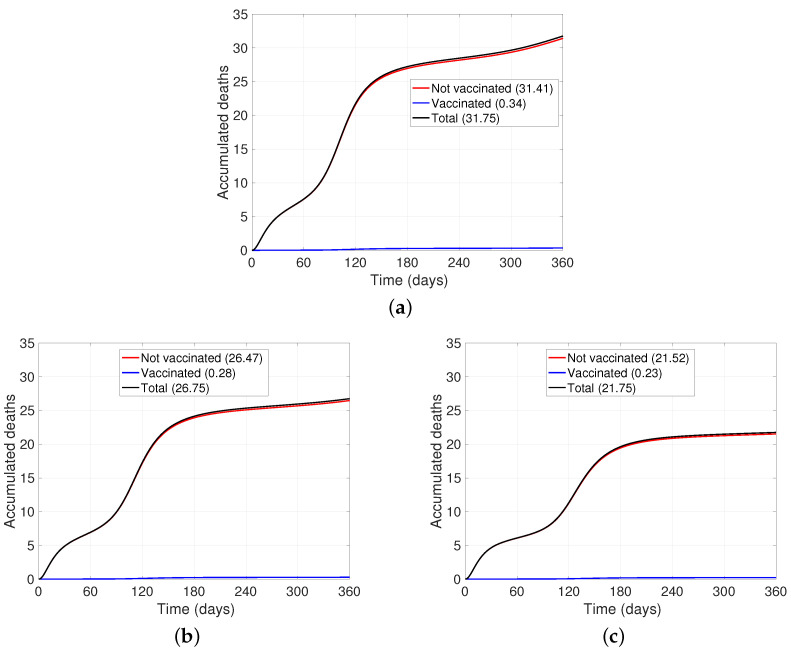
Accumulated deaths associated with SARS-CoV-2. (**a**) Base case. (**b**) Half the rate of resistance to change ($\Lambda_{1}$), that is, people are less reticent to change their behavior with PPE and reduce social distancing. A similar graph is obtained after doubling the speed of people’s reaction to the pandemic ($\Lambda_{2}$). (**c**) Reduction of resistance to change by half and increase in reaction speed by double.

**Table 1 ijerph-19-02022-t001:** Notation of compartments associated with the mathematical model (NVac. = Not vaccinated, Vac = Vaccinated).

	Susceptible	Exposed	Asymptomatic	Infected	Treatment	Recovered
NVac.	*S*	*E*	*A*	*I*	*T*	*R*
Vac.	$S_{v}$	$E_{v}$	$A_{v}$	$I_{v}$	$T_{v}$	$R_{v}$

**Table 2 ijerph-19-02022-t002:** Description of parameters and parameter values related to behavior. D = days, UN = unitless. $N=N_{u}+N_{v}$. In order to not discriminate between people with different beliefs and attitudes towards vaccination, we used the same numeric values for both $P^{*}$ and $P_{v}^{*}$. ${}^{*}$ The detail of the referenced values is presented in the Appendix A, Appendix A.

Parameters	Description	Units
*b* (*d*)	Birth (Mortality) rate	D${}^{-1}$
*f*	Vaccination rate	D${}^{-1}$
$f_{v}$	Loss of immunity of vaccinated	D${}^{-1}$
$\beta_{a}^{*}$ ($\beta_{i}^{*}$)	Transmission rate of asymptomatic (infectious)	D${}^{-1}$
σ (δ)	Susceptibility (Infectivity) reduction factor	UN
α ($\alpha_{v}$)	Exit rate from latent unvaccinated (vaccinated)	D${}^{-1}$
	to infectious unvaccinated (vaccinated)	
$\lambda_{a}$ ($\lambda_{a}^{v}$)	Proportion of latent unvaccinated (vaccinated)	UN
	that transit to asymptomatic unvaccinated	
	(vaccinated)	
$\lambda_{i}$ ($\lambda_{i}^{v}$)	Proportion of latent unvaccinated (vaccinated)	UN
	that transit to infectious unvaccinated	
	(vaccinated)	
ϕ ($\phi_{v}$)	Transition rate from asymptomatic unvaccinated	D${}^{-1}$
	(vaccinated) to infectious unvaccinated (vaccinated)	
μ ($\mu_{v}$)	Transition rate of people unvaccinated (vaccinated)	D${}^{-1}$
	needing medical intervention	
*m* ($m_{v}$)	Disease induced death rate of unvaccinated (vaccinated)	D${}^{-1}$
$\gamma_{a}$ ($\gamma_{a}^{v}$)	Recovered rate of asymptomatic unvaccinated (vaccinated)	D${}^{-1}$
$\gamma_{i}$ ($\gamma_{i}^{v}$)	Recovered rate of infectious unvaccinated (vaccinated)	D${}^{-1}$
$\gamma_{T}$ ($\gamma_{T}^{v}$)	Recovery rate from treatment of people	D${}^{-1}$
	unvaccinated (vaccinated)	
ψ ($\psi_{v}$)	Natural immunity loss rate of people	D${}^{-1}$
	unvaccinated (vaccinated)	
$\Lambda_{1}$ ($\Lambda_{1}^{v}$)	Rate of resistance to behavioral change those	D${}^{-1}$
	unvaccinated (vaccinated)	
$\Lambda_{2}$ ($\Lambda_{2}^{v}$)	Reaction rate to behavior change those	D${}^{-1}$
	unvaccinated (vaccinated)	
$P^{*}$ ($P_{v}^{*}$)	quantified average risk perception of those	UN
	unvaccinated (vaccinated)	
$N_{u}$ ($N_{v}$)	Unvaccinated (vaccinated) population	UN

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
