# Peer review of "Risk Perception Influence on Vaccination Program on COVID-19 in Chile: A Mathematical Model"

_ijerph, 2022, doi:10.3390/ijerph19042022_

Round 1
Reviewer 1 Report
I agree with the modifications made
I think that it can be published in the current state
Author Response
Thank you for your feedback!
Reviewer 2 Report
The authors have answered my question to my satisfaction. no more comments are added in this review.
Author Response
Thank you for your feedback!
This manuscript is a resubmission of an earlier submission. The following is a list of the peer review reports and author responses from that submission.
Round 1
Reviewer 1 Report
The Reader must intuitively guess what the S and R variables represent.
Have the Authors attempted to estimate (simulating) the time of a significant reduction in the incidence? The model seems to be 'expiring' in the long term.
Author Response
We would like to thank Reviewer 1 for his/her suggestions.
1) The Reader must intuitively guess what the S and R variables represent.
R: We appreciate your observation, as we did not realize we did not specify them in the text. We have explicitly mentioned the representation of S and R in the manuscript (p. 3 line 106). As we changed many parts due to the other reviewer’s suggestions, the text of the modified parts throughout the manuscript is in red for a faster review.
2) Have the Authors attempted to estimate (simulating) the time of a significant reduction in the incidence? The model seems to be 'expiring' in the long term.
R: Thank you for observing this. Indeed, when we first created our model, we actually estimated what happens “on the long term” (~3 years), observing that the waves keep on appearing, but slowly decreasing their incidence and amplitude (which is the desired behavior of any pandemics, anyway). Nevertheless, in the present model we were interested in what happened the first year, because we are aware of the important topic (not addressed in this model) of variants depending on the virus mutations, so it was useless to model a longer time, assuming that new variants would change the contagion rate and the other related variables (including effectiveness of vaccines, as it seems it’s happening now with the Omicron variant).
Reviewer 2 Report
This is an interesting study that analyzes human behavior as a key element in the spread of infection.
The study is based on the development of a mathematical model applied to Chilean society to evaluate the interaction between medical factors such as vaccination and personal factors or the perceived risk of becoming infected and the speed of adopting changes aimed at preventing the disease.
I think it is an interesting work and may help to try to translate into numbers the consequences of negative attitudes towards the control of COVID-19.
Introduction
I believe it is excessively long, so it is suggested to reduce the initial paragraphs to focus on the aspects more related to the content of the article.
Author Response
1) This is an interesting study that analyzes human behavior as a key element in the spread of infection.
The study is based on the development of a mathematical model applied to Chilean society to evaluate the interaction between medical factors such as vaccination and personal factors or the perceived risk of becoming infected and the speed of adopting changes aimed at preventing the disease.
I think it is an interesting work and may help to try to translate into numbers the consequences of negative attitudes towards the control of COVID-19.
R: Many thanks for your appreciation of our work.
2) Introduction: I believe it is excessively long, so it is suggested to reduce the initial paragraphs to focus on the aspects more related to the content of the article.
R: Thank you for your suggestion, we rephrased many sentences and “cut” some parts to make it shorter and we marked in red the text of the modified parts in order to make it easier to review.
Reviewer 3 Report
The authors present an interesting use of the Risk Perception problem when dealing with a mathematical model for COVID-19.
Some part of the text needs to be improved; a few examples are provided:
“This part of information might modulate the outcome of many models, bringing them closer to what happens in real life and giving more realistic information about pandemics.”
- “and giving”
“But how can the contagion waves and increase be explained”
- Probably rewriting the phrase would improve readability
“(more information on this topic [24].”
- The parenthesis is missing
Citation to MATLAB and packages use are needed
“The numerical simulations associated with the proposed model (1) were performed using Matlab software.”
Figure 1, the description could be improved if an explanation is presented of the diagram as part of the text for figure 1 and the equation are presented in the text. Please review the equation presented in figure 1 an if the normalization factors of vaccinated and unvaccinated are correct.
There are some important parts on the introduction that would need a citation to provide information to the readers, for example:
“This is a "modern times fear", since many influential people (including health professionals, doctors and scholars) in the lasts 20 years started to spread arguments against vaccines, and the public opinion showed to be highly susceptible and prone to agree with these attitudes.”
Citation work of the studies focus on this behaviour are needed to have a better context of the problem
For the overall work presented there are considerations the author needs to address,
- Figure 2, shows a model where vaccination seems to be considered from the beginning and the waves are dependent on this factor, in a model where risk perception is important, is not clear how not considering access to vaccine (or the lack of) is something that doesn’t need to be consider. Furthermore, the behavior of the two populations seems to be very similar, please show the values of your N = Nu + Nv, or show each population infected with respect to each one total population.
- Figure 3 shows what the author presents treatment with respect to daily doses, the authors claim a change on the peak of the wave due to increase daily doses, but the peaks are also higher, the modification on the unvaccinated in the first 30 days of the models also need to be explained form the base model to the one with increase doses. Is important to show a reason for treatment rate increase with higher vaccination doses in the first 120 days. The explanation of the mixture of the two waves is not clear, due to the impact of vaccination on the total number of treatment and not only as a factor that mix waves
- Figure 4 and 5, Risk perception is a complex descriptor that most of the time consider various social behaviour, feeling with respect to the problem and knowledge of the population. In the presented model it seems that the way the authors present their model only reduce the scale of the death/treatments, but not the behaviour of the pandemic, this is a problem when comparing with information across the world of different societies and their perception of the pandemic and the changes on behaviour.
When reading what the authors are proposing as a novelty to the models, introducing risk perception as a factor, is not clear that the resulting model is describing risk perception and is at this point only modelling to variables that the authors present as an equivalent of risk perception.
Rate of resistance to behavioural change and reaction rate to behaviour change, could be consider as resulting variables of perception, but at this point with what the authors present of the paper is not clear if the values used are related to what is been observed when describing risk perception In the COVID pandemic
Author Response
We thank you for your thorough review of the manuscript. Please see the attachment to see a point-by-point response to all your observations.
